# Potential of *Acidithiobacillus ferrooxidans* to Grow on and Bioleach Metals from Mars and Lunar Regolith Simulants under Simulated Microgravity Conditions

**DOI:** 10.3390/microorganisms9122416

**Published:** 2021-11-23

**Authors:** Anna H. Kaksonen, Xiao Deng, Christina Morris, Himel Nahreen Khaleque, Luis Zea, Yosephine Gumulya

**Affiliations:** 1Commonwealth Scientific and Industrial Research Organisation (CSIRO), Land and Water, Floreat 6014, Australia; anna.kaksonen@csiro.au (A.H.K.); deng.xiao@nims.go.jp (X.D.); christina.morris@csiro.au (C.M.); himelnahreen.khaleque@csiro.au (H.N.K.); 2School of Biomedical Sciences, University of Western Australia, Crawley 6009, Australia; 3International Center for Materials Nanoarchitectonics, National Institute for Materials Science, Tsukuba 305-0044, Japan; 4BioServe Space Technologies, Smead Aerospace Engineering Sciences Department, University of Colorado Boulder, Boulder, CO 80303, USA; Luis.Zea@Colorado.edu; 5Centre for Microbiome Research, School of Biomedical Sciences, Translational Research Institute, Queensland University of Technology, Woolloongabba 4102, Australia

**Keywords:** *Acidithiobacillus ferrooxidans*, bioleaching, microgravity, lunar and Mars regolith, space mining

## Abstract

The biomining microbes which extract metals from ores that have been applied in mining processes worldwide hold potential for harnessing space resources. Their cell growth and ability to extract metals from extraterrestrial minerals under microgravity environments, however, remains largely unknown. The present study used the model biomining bacterium *Acidithiobacillus ferrooxidans* to extract metals from lunar and Martian regolith simulants cultivated in a rotating clinostat with matched controls grown under the influence of terrestrial gravity. Analyses included assessments of final cell count, size, morphology, and soluble metal concentrations. Under Earth gravity, with the addition of Fe^3+^ and H_2_/CO_2_, *A. ferrooxidans* grew in the presence of regolith simulants to a final cell density comparable to controls without regoliths. The simulated microgravity appeared to enable cells to grow to a higher cell density in the presence of lunar regolith simulants. Clinostat cultures of *A. ferrooxidans* solubilised higher amounts of Si, Mn and Mg from lunar and Martian regolith simulants than abiotic controls. Electron microscopy observations revealed that microgravity stimulated the biosynthesis of intracellular nanoparticles (most likely magnetite) in anaerobically grown *A. ferrooxidans* cells. These results suggested that *A. ferrooxidans* has the potential for metal bioleaching and the production of useful nanoparticles in space.

## 1. Introduction

In situ resource utilization (ISRU), defined as the use of local resources to produce space consumables, is increasingly gaining importance with long-duration missions such as those to Mars, as the resupply from Earth is cost-limiting. The recent launch of NASA’s Moon to Mars program, aiming to lay the foundation for a sustained long-term human presence on the lunar surface will also require ISRU technologies that enable the production of commodities using natural resources from the Moon or other celestial bodies. Robust microorganisms can be used and potentially engineered using synthetic biology approaches as micro-factories for transforming in situ destination planet resources into useful products. It is known that microbes interact with minerals in the rocks or soils on Earth, contributing to the geochemical cycling of elements. Several chemolithoautotrophic microorganisms can dissolve valuable metals from minerals and wastes in the process called biomining [1,2]. Biomining has been successfully employed on Earth to extract copper, nickel, cobalt, zinc and uranium [3]. The use of microbes to extract resources from regoliths to establish space infrastructure can help tackle some economic and logistic challenges as they can be transported in small quantities and allowed to reproduce inside an extraterrestrial bioreactor. Biomining microbes could additionally be used as a microbial chassis to produce feedstocks for manufacturing processes from space junk, nutrients for astronauts, and on-demand space medicine, hence assisting in addressing the grand challenges in space synthetic biology [4].

The idea of using microorganisms to extract resources from regoliths have been explored as early as 2007 with NASA’s workshop on lunar regolith biomining. It was suggested that microbes might be able to extract metals and other resources from lunar materials, given suitable growth conditions [5]. The discovery of sulfate minerals in the Meridiani Planum on Mars [6,7] that indicates past aqueous acidic conditions led to a hypothesis for the role of chemolithoautotrophic microorganisms in the formation of these deposits. Several studies examining the ability of biomining microbes to grow on the minerals contained in space regoliths or assessing their survival capability when exposed to simulated space conditions have subsequently been reported [8,9,10]. For instance, *Acidithiobacillus ferrooxidans* was able to grow on the minerals contained in synthetic Mars regolith aerobically, using O_2_ as an electron acceptor for iron oxidation, or anaerobically, using H_2_ as an electron donor for iron reduction [8]. *A. ferrooxidans* could also oxidize iron from meteorites (Casas Grandes and Cape York) without the addition of extra nutrients [9]. Under the protection of a thin layer of ferric oxides and hydroxides, *A. ferrooxidans* could survive after being exposed for 10 h to simulated Mars environmental conditions [10]. Although microbes were able to survive in simulated space conditions, the poly-extreme nature of space will necessitate the use of containment when using terrestrial microbes in space. Recently, a prototype extraterrestrial biomining reactor was developed, and an International Space Station experiment, Biorock, was carried out with this prototype for studying microbe mineral interactions in space [11]. In addition to chemolithoautotrophs, dissimilatory metal-reducing bacteria such as *Shewanella oneidensis* that can respire anaerobically using solid metal oxides as a terminal electron acceptor and convert the oxides to more soluble forms have also been investigated. The theoretical feasibility (process yield and kinetics) of using *A. ferrooxidans* and *S. oneidensis* to extract iron from lunar and Martian regolith were simulated [12]. The estimated payback time of the infrastructure installation for the bioleaching process using *S. oneidensis* on Mars in this study was 3.3 years, with 44.7 g/L of iron being produced from 300 g/L Martian regolith [12].

Terrestrial microbes grown inside a space bioreactor will be exposed to different gravity conditions (e.g., lunar/Martian gravity, microgravity or weightlessness) depending on the location of the reactor. In addition to the reduction of the physical force of gravity, the microbes will encounter changes in the fluid dynamics or shear forces that occur during the microgravity of spaceflight and during the ground-based microgravity analogue. Several studies reported increased pathogenicity, enhanced biofilm formation or improved tolerance to antibiotics and other extreme conditions (e.g., osmolarity, pH, temperature) as microbial responses to microgravity and other low-shear environments [13,14]. Assessing the leaching performance of biomining microbes in ground-based microgravity-simulating devices could be important for developing biomining technology for space applications. *A. ferrooxidans* and *Metallosphaera sedula* were able to release free soluble metals into the medium when they were grown in regolith simulants [15,16]. No study has previously measured the effect of microgravity on the metal solubilization of lunar and Martian regolith simulants using chemolithoautotrophs. So far, only the heterotroph *S. oneidensis* MR-1 has been studied for its ability to reduce iron under space flight conditions [17]. Here, the same space flight hardware: the fluid processing apparatus (FPA) and group activation pack (GAP), used in the space flight experiment with *S. oneidensis*, was used to explore the growth potential and metal extraction capability of the biomining microbe *A. ferrooxidans* under a simulated microgravity environment. The *A. ferrooxidans* was cultivated in the presence of lunar and Martian regolith simulants under anaerobic conditions using H_2_ as the electron donor. The microbial formation of magnetosomes under simulated microgravity was also evaluated.

## 2. Materials and Methods

### 2.1. Bacterial Strain, Media and Growth Conditions

*A. ferrooxidans* DSM 14882^T^ inoculum (10%) was grown in 125 mL serum flasks under anaerobic conditions for 7 days in 50 mL of modified 9K medium, which contained (g L^−1^): 0.4 (NH_4_)_2_SO_4_, 0.4 MgSO_4_·7H_2_O, 0.4 KH_2_PO_4_ and 1 mL of trace element solution. The trace element solution contained (mg L^−1^): 62 MnCl_2_·2H_2_O, 68 ZnSO_4_·7H_2_O, 64 CoCl_2_·6H_2_O, 30 H_3_BO_3_, 10 Na_2_MoO_4_, 66 CuCl_2_·2H_2_O and 30 NaVO_3_. The pH was adjusted to 1.8 with concentrated H_2_SO_4_. The medium was made anoxic by purging with N_2_ gas for 30 min. The medium was supplemented with 2.5 g L^−1^ Fe_2_(SO_4_)_3_ as the electron acceptor, while 2.50 mmol H_2_ was supplied as the electron donor by filling the 75 mL headspace with an H_2_:CO_2_ (80:20, *v/v*) gas mix using 0.8 bar overpressure.

### 2.2. Composition of Lunar and Martian Regolith Simulants

In this study, two regolith simulants, LMS-1 (lunar mare simulant) and MGS-1 (Mars global simulant) were used to examine whether these mineral mixtures could provide nutrients or energy necessary for lithoautotrophic growth of *A. ferrooxidans*. The regolith simulants were purchased from Exolith Lab (https://sciences.ucf.edu/class/exolithlab/, accessed on 12 March 2019), and their chemical and mineral compositions were as listed in Table 1 and Table 2, respectively.

### 2.3. Microgravity Simulation Experiment

A BioServe Space Technologies GAP clinostat and FPA space flight hardware (previously operated in space over 5000 times) were used to grow *A. ferrooxidans* under a simulated microgravity (10^−6^× *g*) environment via clinorotation at 8 rpm [18,19]. LMS−1 and MGS-1 regolith simulants were sterilized by tyndallization at 100 °C for one hour over three successive days. After adding 13.5 mL of anoxic modified 9K medium into the fluid processing apparatus (FPA; BioServe, Boulder, CO, USA), 1% *w*/*v* of sterilized regolith and a 10% *v/v* 7-day inoculum of *A. ferrooxidans* culture were introduced into the FPA inside an anaerobic bag (Aldrich^®^ AtmosBag, Merck, NJ, USA) filled with an H_2_:CO_2_ (80:20, *v/v*) gas mix. The initial cell number in each FPA was approximately 10^7^ cells/mL. Some FPAs were supplemented with 2.5 g L^−1^ Fe_2_(SO_4_)_3_ (i.e., 12.5 mM Fe^3+^) as the electron acceptor (Table 3). Six FPAs were assembled into a group activation pack (GAP; BioServe, Boulder, CO, USA) (Figure 1A) and tightly sealed inside the anaerobic bag. The 18.2 mmoL of H_2_ in the 547 mL headspace of the GAP can diffuse into FPAs via gas-exchange (GE) membranes in each end and served as the electron donors for the cells. The cultures were incubated for 20 days at 30 °C under microgravity or Earth gravity conditions, which were maintained by rotating the GAP on a clinostat or letting it stand vertically, respectively (Figure 1B). Uninoculated FPAs in a GAP on Earth gravity conditions were used as abiotic controls (Table 3).

### 2.4. Physicochemical Analysis

After the 20-day incubation, samples (1 mL) were filtered through a 0.2 µm syringe filter (Terumo, Philippines) and collected in 1.5 mL tubes. The filtered samples were then tested for physicochemical parameters such as pH, redox potential and metal concentrations, as follows. Ferric iron concentrations were measured using modifications to the method previously described [20]. In brief, ferric iron standards (0–500 mg L^−1^) were prepared from a standard solution (1000 mg L^−1^ Fe(NO_3_)_3_ in 7 M HNO_3_; Merck). The indicator solution comprised 2 M HCl made using ultrapure water from Elga Purelab Flex water purifier. Sample and HCl solutions were combined in a volumetric ratio of 1:1 and mixed thoroughly. Modified 9K medium was used for blanks. Absorbance was measured at 340 nm in the Varioskan Lux plate reader (Thermo Scientific, Parkville, Australia) after loading samples into 96-well plates (Sarstedt, Mawson Lakes, Australia) in triplicate. Solution pH and redox were measured using the S2 Seven2Go™ meter and InLab^®^ pH and redox microelectrodes, respectively. Redox potentials were reported against an Ag/AgCl reference. Samples for inductively coupled plasma–optical emission spectrometry (ICP-OES) were prepared by adding 1 mL of filtered sample to 0.1 mL 7M HNO_3_ and 8.9 mL ultrapure water. Where necessary, the samples were further diluted with Hamilton autodiluter and analyzed using a Varian 730-ES ICP-OES.

### 2.5. Cell Counting

Cell counting was carried out for each inoculated FPA. FPA was mixed well by shaking before sampling, and 100 µL aliquots were transferred to a 1.5 mL microcentrifuge tube. The tubes were sonicated three times for 30 s with a 30 s rest period between each sonication to detach and separate the bacteria from minerals. The regolith simulant in the tubes was left to settle for 30 min before taking the supernatant for measurement. Total cell count was performed in duplicate using a Helber Bacterial Counting Chamber SV400.

### 2.6. Scanning Electron Microscopy (SEM)

After incubating the cells under simulated microgravity and Earth gravity conditions for 20 days, 2 mL aliquots of cultures were immediately fixed with 2 mL of 5% glutaraldehyde prepared in phosphate buffer (0.1 M, pH 7) for approximately 16 h and pelleted by centrifugation at 10,000× *g* for 10 min. The fixed pellets were resuspended in 50 µL phosphate buffer (0.1 M, pH 7) and transferred to the surface of glass plates precoated with 0.01% (*w*/*v*) poly-l-lysine and incubated for 30 min to allow cell attachment. The glass plates with attached cells were washed three times with phosphate buffer (0.1 M, pH 7) and dehydrated in gradients of 25, 50, 75, 90, and 100% ethanol. The 100% ethanol was exchanged twice before the samples were freeze-dried in a Polaron KE3000 critical point drier (Quorum, Lewes, UK). The samples were then sputter-coated with Pt and viewed using a secondary electron imaging mode with a Verios XHR SEM (FEI, Hillsboro, OR, USA).

### 2.7. Scanning Transmission Electron Microscopy (STEM)

After the incubation experiments, the cells were fixed and dehydrated as described above for SEM. The dehydrated cells in 100% ethanol were pelleted by centrifuging at 10,000× *g* for 10 min and were then infiltrated with 1 mL of 50% LR-white resin in 100% ethanol for 30 min at room temperature, and with 1 mL of 100% LR-white resin for 1 h at room temperature. After infiltration, cells were pelleted by centrifugation at 10,000× *g* for 10 min, and 0.5 mL fresh 100% LR-white resin was added to the pellet and resuspended gently without forming air bubbles. The resin was polymerized at 60 °C for 2 days and cut into 80 nm-thick sections by a diamond knife (DiATOME, Hatfield, PA, USA). The sections were attached onto carbon-coated copper microgrids and viewed using a STEM mode with a Verios XHR SEM (FEI, Hillsboro, OR, USA).

### 2.8. Statistical Analysis

The statistical significances of the differences in results obtained for cultures incubated under various conditions were determined via unpaired two-sided *t*-tests, using R.

## 3. Results

The ability of *A. ferrooxidans* to extract metals from lunar and Martian regolith simulants was evaluated under simulated microgravity and Earth gravity conditions. *A. ferrooxidans* cultures were grown anaerobically using H_2_ as an electron donor inside FPAs with four substrate alternatives and two gravitational conditions (Figure 2).

### 3.1. Cell Growth

*A. ferrooxidans* appeared not to be able to grow solely in the presence of mars regolith simulants (MR), without the addition of Fe_2_(SO_4_)_3_ (Figure 3). The cell density in MR cultures remained similar to initial cultivation. The absence of observable growth in MR cultures was likely because the carbon source (CO_2_) provided in the gas mixture or the electron donor (Fe^3+^) gained from the regolith simulants was not sufficient to support growth, possibly leading the cells to starvation [21]. Under Earth gravity, the growth of *A. ferrooxidans* in MRS cultures was enhanced when cultivated with the Mars regolith simulant plus Fe_2_(SO_4_)_3_ (MRS vs. S in Figure 3, *p*-value = 0.0007). This was likely because MRS contains a high amount of iron (in ferrous and ferric forms) (Table 1).

Simulated microgravity affected the growth of *A. ferrooxidans* differently depending on which regolith simulants were added into the cultures. The cell density of the S and LRS cultures grown in clinostat appeared to be slightly higher than in Earth gravity. However, the difference between them was not statistically significant (*p*-value = 0.0795 and 0.2765 for S and LRS cultures, respectively). The number of cells in the MRS clinostat cultures were significantly lower (*p*-value = 0.0028) than in control cultures. The extra energy input presented in MRS may not be sufficient to encounter the stress caused by the regolith simulant and simulated microgravity conditions. The low cell counts in clinostat MRS cultures did not reflect the high ferric consumption rate presented in Table 4. Except for MR that did not get supplementation of Fe_2_(SO_4_)_3_, all cultures showed a higher ferric consumption rate when cultivated under simulated microgravity than in Earth gravity.

### 3.2. Changes in Ferric and Total Soluble Iron Concentration, Redox Potential and pH

The soluble ferric and total iron concentrations, redox potential and solution pH in the FPAs at the start and after 20 days of incubation are shown in Figure 4. The ferric iron concentration results for the substrate only (S), as well as both Mars and lunar regolith simulant cultures supplemented with Fe_2_(SO_4_)_3_ (MRS and LRS) showed a decrease after 20 days of incubation (Figure 4A). Together with the relatively minor decrease in total soluble iron concentrations (Figure 4B), this suggested ferric iron reduction and, hence, microbial activity. In cultures incubated under both simulated microgravity and Earth gravity conditions, there appeared to be lower ferric consumption with Fe^3+^ only (S) cultures as compared to the regolith cultures amended with Fe^3+^ (LRS and MRS) (Figure 4A and Table 4). This indicated that the regolith simulant did not have an inhibiting effect on anaerobic ferric iron reduction. No iron reduction was detected in the cultures with regolith simulant (MR-IE and MR-IM) only or in the abiotic tubes with substrate only (S-AE) or with regolith simulant only (MR-AE) (Figure 4A). However, some ferric iron reduction was detected in the abiotic tubes amended with both regoliths and substrate (MRS and LRS), although to a lesser extent than in the inoculated tubes, indicating possible ferric precipitation. Interestingly, simulated microgravity appeared to enhance the ferric reduction in all cultures amended with Fe^3+^ (S, MRS and LRS). The increase in ferric consumption was significantly higher in the S and MRS cultures grown under simulated microgravity than under Earth gravity (*p*-value = 0.0031 and 0.0003, respectively, Table 4).

In the presence of Mars regolith (MR), the total soluble iron concentration in inoculated FPAs increased under microgravity conditions as compared to time 0 and incubation under Earth gravity conditions (Figure 4B). This was consistent with the lower redox potential in the MR culture incubated under microgravity conditions as compared to under Earth gravity conditions. However, abiotic FPAs with the Mars regolith simulant (MR) showed a higher increase in soluble Fe than inoculated FPAs (*p*-value = 0.0002). In the Mars and lunar regolith simulant cultures supplemented with Fe^3+^ (i.e., MRS and LRS cultures, respectively) and incubated under Earth gravity or simulated microgravity conditions, soluble Fe concentrations were lower after 20 days of incubation as compared to the start of the experiment. A similar decrease in soluble Fe was detected in abiotic FPAs over time, suggesting iron precipitation.

All cultures, including MR, had lower redox potential after 20 days of incubation than at the start of the experiment, suggesting that reduction of Fe^3+^ to Fe^2+^ occurred (Figure 4C), which was consistent with the Fe^3+^ and total dissolved Fe results (Figure 4A,B). Redox potential also decreased somewhat in the abiotic FPAs that contained regolith simulants, while in the abiotic Fe^3+^ only FPA, no notable decrease in redox potential was detected. The solution pH increased slightly over time in all regolith simulant-containing FPAs, including in the abiotic controls. In inoculated FPAs, the pH increase was, however, more notable (Figure 4D). The pH increase in the abiotic FPAs (from pH 2 to ~2.5–3) was likely caused by acid consuming minerals in the regolith simulants. In regolith simulant-containing cultures, the pH increase was higher under simulated microgravity conditions than under Earth gravity.

### 3.3. Solubilisation Elements under Earth Gravity and Simulated Microgravity Conditions

The effect of simulated microgravity on the solubilization of elements from lunar and Mars regolith simulants was evaluated by analyzing the soluble concentrations of the selected elements after the 20-day incubation under Earth gravity (control) and simulated microgravity conditions. The regolith elements that were considered valuable for mining and were analyzed in this study were iron, aluminum (for space construction materials) and silicon (for solar cells). Other elements analyzed were manganese, sulfur and titanium. Solubilization of essential cations for life such as sodium, potassium, calcium and magnesium were also investigated. Simulated microgravity had different impacts on the leaching of various elements, as shown in Figure 4B, Figure 5 and Figure 6.

The leaching of aluminum from Martian and lunar regolith simulants was higher in abiotic FPAs than in inoculated FPAs (*p*-value < 0.0001) (Figure 5A). In inoculated FPAs, Al leaching was somewhat higher under Earth gravity conditions than under simulated microgravity. Silicon leached readily from both Mars and lunar regolith simulants in both inoculated and abiotic FPAs (Figure 5B). In FPAs with Mars regolith simulant only (MR), abiotic FPAs showed higher leaching than inoculated FPAs (*p*-value < 0.0001). On the other hand, inoculated FPAs containing Mars or lunar regolith simulants and Fe^3+^ (MRS and LRS) showed higher leaching when incubated under simulated microgravity conditions than the inoculated FPAs under Earth gravity or the abiotic FPAs (*p*-value < 0.0001 for MRS and *p*-value = 0.0003 for LRS). The leaching of Si from the Mars and lunar regolith simulant in the presence of Fe^3+^ was similar in the inoculated and abiotic FPAs under Earth gravity conditions. No titanium leaching was detected under any conditions, as the soluble titanium concentrations were lower after 20 days than on day 0 for all regolith simulant-containing FPAs (Figure 5C).

The amount of manganese leached from Mars regolith simulants without substrate amendment (MR) in inoculated FPAs was slightly higher than that detected in abiotic FPAs (Figure 5E). On the other hand, for FPAs amended with Fe^3+^ (MRS and LRS), the inoculated FPAs incubated under both Earth gravity and simulated microgravity showed notably higher Mn leaching than the abiotic FPAs (*p*-value = 0.0008 for MRS and *p*-value = 0.0007 for LRS). Moreover, the simulated microgravity seemed to increase Mn leaching from MRS in inoculated FPAs supplemented with Fe^3+^. Soluble sulfur concentration increased over time in all FPAs (Figure 5F). In inoculated FPAs with Mars regolith simulant and Fe^3+^ (MRS), the increase in soluble sulfur concentration was similar to that detected in abiotic FPAs. On the other hand, FPAs with Mars regolith simulant only (MR) or lunar regolith simulant and Fe^3+^ (LRS), the increase in soluble sulfur concentration was higher in inoculated FPAs than in abiotic FPAs (*p*-value = 0.0003 for MR and *p*-value < 0.0001 for LRS). The concentrations were similar for inoculated FPAs incubated under both Earth gravity and simulated microgravity conditions.

When comparing to day 0, the leaching of Na from Mars regolith simulant without Fe^3+^ (MR) in inoculated FPAs was somewhat lower than in abiotic FPAs, but the difference was not statistically significant (Figure 6A, *p*-value = 0.1073). For FPAs containing Mars regolith simulant and Fe^3+^ (MRS), the increase in soluble Na concentration was higher in the inoculated FPAs under microgravity and abiotic FPAs than in the inoculated FPAs under Earth gravity conditions (*p*-value = 0.007). For FPAs with lunar regolith simulant and Fe^3+^ (LRS), cultures incubated under microgravity showed more Na leaching than abiotic FPAs. However, the inoculated FPAs under Earth gravity had lower Na leaching than abiotic FPAs. Soluble potassium concentration decreased over time in all inoculated FPAs under both Earth gravity and simulated microgravity conditions, whereas abiotic FPAs with regolith simulants (MR, MRS and LRS) showed a slight increase in K concentrations over time (Figure 6B).

Magnesium leached in all FPAs (Figure 6C). For FPAs with Mars regolith simulant (MR), Mg leaching was lower in inoculated FPAs than in abiotic FPAs (*p*-value < 0.0001), whereas for FPAs with Mars regolith simulant and Fe^3+^(MRS), inoculated FPAs showed higher leaching than abiotic ones. Moreover, for the inoculated MRS FPAs, the Mg leaching was higher under simulated microgravity than under Earth gravity conditions (*p*-value < 0.0001). For FPAs with lunar regolith simulant and Fe^3+^, the inoculated cultures incubated under microgravity showed also higher leaching than those under Earth gravity or abiotic FPAs (*p*-value = 0.0026). Calcium leached more from Mars regolith simulant than from lunar regolith simulant (Figure 6D). For Mars regolith simulant without Fe^3+^ (MR), the leaching was similar in the inoculated and abiotic FPAs. For FPAs with Mars regolith simulant and Fe^3+^ (MRS), the abiotic FPAs showed higher Ca leaching than the inoculated FPAs (*p*-value = 0.0181). The leaching of Ca in inoculated FPAs with lunar regolith simulant and substrate (LRS) under Earth gravity conditions was lower than that under simulated microgravity conditions and in abiotic FPAs (*p*-value < 0.0001).

### 3.4. SEM and STEM Observation of Cell Morphology

SEM analyses were conducted for *A. ferrooxidans* after 20-day cultivation under Earth gravity and simulated microgravity conditions to examine possible changes in cell morphology (Figure 7). No significant difference in the cell size was observed between the two conditions (Appendix A). The lack of changes in the thickness and structure of the cell envelope was further confirmed by STEM observation of 80-nm thick cell sections (Figure 8). This may be because motile *A. ferrooxidans* potentially disrupted the quiescent extracellular environment by active cell movement. However, STEM revealed that under simulated microgravity conditions, *A. ferrooxidans* cells produced spherical mineral nanoparticles in the cytoplasm. Similar spherical nanoparticles could not be observed in the cytoplasm of the cells cultured under Earth gravity.

## 4. Discussion

In the next decade, space exploration will likely move beyond low Earth orbit into deeper space. A long journey through the solar system would require novel means for providing supplies that rely on in situ destination resources. Robust organisms could be engineered to produce on-demand space consumables using synthetic biology approaches. Chemolithoautotrophs such as *A. ferrooxidans* are emerging space synthetic biology hosts, given their ability to grow on minerals contained in the regolith [8,9]. Albeit less explored, *A. ferrooxidans* can also synthesize intracellular magnetite magnetosomes [22], which have recently drawn increasing interest as nanoscale drug carriers [23]. Hence, acidophiles such as *A. ferrooxidans* can be viewed as micro-factories capable of providing supplies for long-term missions such as construction metals (e.g., leaching of minerals contained in regolith) and iron oxide nanoparticles (e.g., magnetosome). To be used as synthetic biology host organisms, the growth, survival and adaptability of *A. ferrooxidans* in harsh space environments needs to be evaluated. While using a purposely-designed biomining reactor has addressed the issue of growing terrestrial microorganisms in a harsh space environment [11], the microbes still experience microgravity throughout the journey. Here, we evaluated the ability of *A. ferrooxidans* to grow, leach metals and form magnetosomes under simulated microgravity conditions.

*A. ferrooxidans* was previously shown able to grow solely on the minerals contained in Mars regolith simulants with no added nutrients [8]. However, we observed no growth in *A. ferrooxidans* cultures when grown without additional Fe^3+^. This is likely due to the amount of electron donor present in MRS-1 regolith simulant being significantly lower (about 10-20 times lower) than in the synthetic regolith used in [8]. Spaceflight and ground-based experiments suggested that microbes showed diverse effects when exposed to (simulated) microgravity [24], which might influence their viability and survival. Microbial growth responses to microgravity and its analogs were mainly dependent on two aspects, the motility of the strains used in the experiments and experimental conditions (e.g., suspension or agar cultures, high or low nutrient concentrations, oxygen availability of the liquid media [24]. Motile cells in liquid cultures can actively agitate the quiescent fluid environment under microgravity, hence reducing the fluid environment differences between microgravity and Earth gravity (control) cultures [25]. The growth of mostly non-motile cells in suspension cultures was shown to exhibit an increased final cell density under microgravity [25,26]. *A. ferrooxidans* has a twitching motility which occurs through type IV pili production [27]. The growth of *A. ferrooxidans* cultures was, however, increased when they are grown in clinostat, indicating that other fluid dynamics parameters (e.g., the absence of sedimentation, convection or hydrostatic pressure under microgravity) might influence the extracellular transport, and hence the cellular growth [24]. For instance, under simulated microgravity, the regolith simulants will be in a free-fall state as the microbes are not settling down at the bottom of the FPAs as they would in terrestrial gravity, potentially providing better access to the cells.

Less cell growth was observed when *A. ferrooxidans* cultures were grown with Mars regolith simulants plus Fe^3+^ in clinostat than in Earth gravity. The cells, however, were metabolically active, as demonstrated by a higher decrease in redox potential (or higher ferric reduction). This could be the adaptation strategy used by the cells in response to the simulated microgravity environment, such as rerouting the energy resources allocated for biomass formation into the expression of stress-related proteins (e.g., DNA repair mechanism). Proteins and genes of numerous stress responses showed altered abundance and expression after exposure to outer space (real) and simulated environments (reviewed in detail in [28]). The regolith simulants additionally exposed the cells to sub-optimal pH, likely due to the presence of acid-consuming minerals in the regolith. The highest pH exposure was experienced by cultures grown in clinostat and without the addition of Fe^3+^. These results suggested that the design of a space biomining bioreactor needs to consider continuous alkaline addition to reduce the abiotic stress encountered by cells and hence increase the leaching performance.

Studies have shown that lunar and Martian regoliths comprise a high amount of silica (SiO_2_) and alumina (Al_2_O_3_) and a moderate amount of calcium oxide (CaO), which could be used for the production of cementitious-based construction materials [29]. Regolith also contains bioessential cations such as Na^+^, K^+^, Mg^2+^, Ca^2+^, which could provide resources to sustain the growth of organisms of a higher taxon. Under terrestrial gravity, *A. ferrooxidans* mobilized higher concentrations of Mg and Mn from Martian and lunar regolith simulants than abiotic ones when cultivated with the addition of Fe^3+^. Higher soluble S was measured in the growth medium of *A. ferrooxidans* cultured in the presence of the lunar regolith simulant plus ferric, and Mars regolith. More metals (Na, K, Mg, Ca, Mn, S and Si) were leached when extreme thermoacidophile *Metallosphaera sedula* cultures were grown with Martian regolith simulants [16]. The drop in K concentrations was detected in leachate solutions of all biotic cultures, indicated its consumption by microbes in order to maintain the growing population. The decrease of total iron in the biotic cultures suggested the formation of insoluble iron oxyhydroxides and hence they were not included in the leachate solution used for ICP-OES analysis. Iron precipitation was also detected in abiotic FPAs over time. It should, however, be noted that these abiotic cultures were grown under Earth gravity conditions, and not under simulated microgravity.

Under simulated microgravity, *A. ferrooxidans* leached a higher amount of Si, Mn and Mg when cultivated in the presence of Mars and lunar regolith simulants plus Fe^3+^ than abiotic controls. A recent International Space Station (ISS) biomining experiment demonstrated that the efficacy of the bioleaching process in the space environment depends on the microbes used, the metals leached and the gravity the applied [30,31]. Bioleaching of rare earth elements (REEs) from basaltic rock using three microorganisms showed that increased leached concentrations of REEs were obtained with *Sphingomonas* (*S.*) *desiccabilis*, but not with *Bacillus subtilis* or *Cupriavidus metallidurans*. The statistical difference between the biological and abiotic leaching processes of REEs was only observed in simulated Mars and Earth gravities, but not in microgravity [30]. *S. desiccabilis* and *B. subtilis* enhanced the leaching of vanadium under all gravities (microgravity, simulated Mars and Earth gravity) compared to sterile controls [31].

In addition to solubilizing metals, *A. ferrooxidans* is capable of synthesizing magnetic nanoparticles, known as magnetosomes. Magnetosomes, membrane-enclosed intracellular crystals of a magnetic iron mineral [32], provides a means for magnetotactic bacteria to navigate in response to Earth’s magnetic field. The synthesis of magnetosomes is favored under low redox potential and under oxygen-limited conditions [33]. The optimum conditions for magnetosome formation in *A. ferrooxidans* were reported for aerobic growth, 160 mM FeSO_4_, 2.4 g/L (NH_4_)_2_SO_4_, 20 °C, pH 1.75 and 50% loadings of medium (which correlated with oxygen concentration) [34]. We demonstrated here that *A. ferrooxidans* was also able to form magnetosomes when grown under anoxic conditions. Furthermore, magnetosome biosynthesis was enhanced when the microbes were grown under simulated microgravity. On the contrary, magnetotactic bacteria *Magnetospirillum magnetotacticum* lost their magnetosome and hence failed to exhibit magnetotaxis when grown aboard the space shuttle and the space station MIR [35]. The chain of magnetite was not formed or not preserved in the microgravity environment, likely correlated with the loss of cellular integrity in *M. magnetotacticum*. Electron micrographs of the ultrathin section of *A. ferrooxidans* BY-3 showed that only 1-3 magnetosomes per cell was found inside the cells, and no chain of magnetosomes was observed [22].

There has been increasing interest in the use of magnetosomes for drug delivery due to their intrinsic properties (e.g., magnetotaxis, low toxicity, biocompatibility, uniform shape and size, good dispersibility under physiological conditions, high surface to volume ratio, easy to modify, etc.) [36]. The abundance of primary amino groups in the surface of magnetosomes can be functionalized with diverse bioactive molecules (e.g., ligands) to potentially target the affected area and hence reduce the adverse side effects. Bacterial magnetosomes, synthesized by magnetotactic bacteria or by *A. ferrooxidans*, could potentially be used for similar applications, considering there is no immediate access to medical support in space. While the biosynthesis of magnetite nanoparticles is well known for cells aerobically cultivated with Fe^2+^/O_2_ [34], it has not been studied for anaerobically cultivated cells. The finding that microgravity stimulated the nanoparticle synthesis in anaerobically grown *A. ferrooxidans* suggests the potential of using *A. ferrooxidans* cells to produce useful nanoparticle materials in space.

## 5. Conclusions

We have demonstrated here that the biomining model organism *A. ferrooxidans* could potentially be used for space applications. *A. ferrooxidans* was able to grow anaerobically and solubilize metals from lunar and Martian regolith simulants, which mainly consist of oxides, under simulated microgravity conditions. Depending on the metals, a different performance of leaching was observed in comparison to the abiotic (acid) leaching. The biosynthesis of intracellular magnetite was additionally increased under simulated microgravity, thus opening the possibility of using *A. ferrooxidans* to produce nanoparticles in space. Furthermore, as deep space missions will go beyond low Earth orbit, the long-term exposures of astronauts to microgravity will require novel means of drug delivery (e.g., using nanoparticles).

## Figures and Tables

**Figure 1 microorganisms-09-02416-f001:**
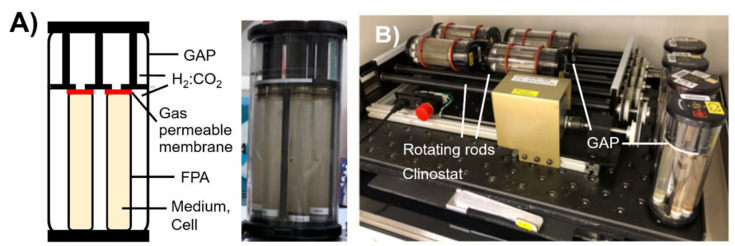
Experimental set-up for incubation of *A. ferrooxidans* under simulated microgravity and Earth gravity conditions. (**A**) Schematic image and a photo of BioServe’s fluid processing apparatus (FPA) and group activation pack (GAP) used for cultivation. (**B**) A photo showing GAPs kept rotating on a clinostat to simulate a microgravity condition and standing vertically to represent the Earth gravity condition.

**Figure 2 microorganisms-09-02416-f002:**
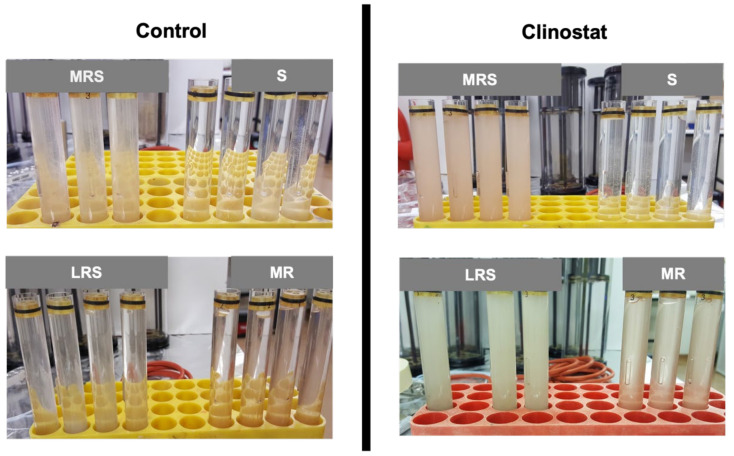
*A. ferrooxidans* cultured in BioServe’s fluid processing apparatus (FPA) with four substrate alternatives (1% Mars regolith simulant plus 12.5 mM Fe^3+^ (MRS); 12.5 mM Fe^3+^ (S); 1% lunar regolith simulant plus 12.5 mM Fe^3+^(LRS); 1% Mars regolith simulant (MR) and two gravitational conditions (control and clinostat) at 30 °C for 20 days. H_2_/CO_2_ gas mixture was added to all cultures as electron donor and carbon sources, respectively.

**Figure 3 microorganisms-09-02416-f003:**
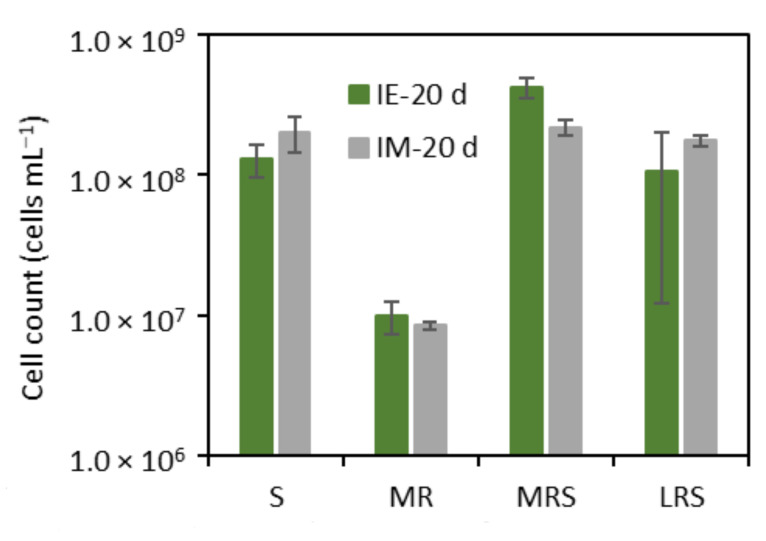
Cell counts of *A. ferrooxidans-*ionoculated (I) cultures after incubation for 20 days under Earth gravity (E) and simulated microgravity (M) conditions in the presence of 12.5 mM Fe^3+^ (S); 1% Mars regolith simulants (MR), 1% Mars regolith simulants plus 12.5 mM Fe^3+^ (MRS) and 1% lunar regolith simulants plus 12.5 mM Fe^3+^ (LRS). H_2_/CO_2_ gas mixture was added to all cultures as electron donor and carbon sources, respectively. The initial cell density was 1 × 10^7^ cells mL^−1^. Error bars show standard deviations.

**Figure 4 microorganisms-09-02416-f004:**
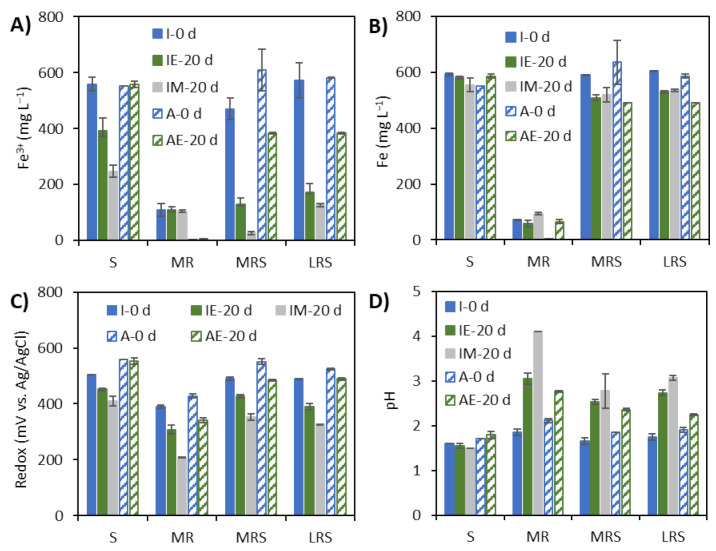
Initial (0 d) and final (20 d) (**A**) ferric iron concentration, (**B**) total iron concentration, (**C**) redox potential and (**D**) pH of *A. ferrooxidans* inoculated (I) cultures and abiotic controls (A) after incubation under Earth gravity (E) and simulated microgravity (M) conditions in the presence of 12.5 mM Fe^3+^ (S), 1% Mars regolith simulant (MR), 1% Mars regolith simulant plus 12.5 mM Fe^3+^ (MRS) and 1% lunar regolith simulant plus 12.5 mM Fe^3^ (LRS). H_2_/CO_2_ gas mixture was added to all cultures as source of electron donor and carbon source, respectively. Error bars show standard deviations.

**Figure 5 microorganisms-09-02416-f005:**
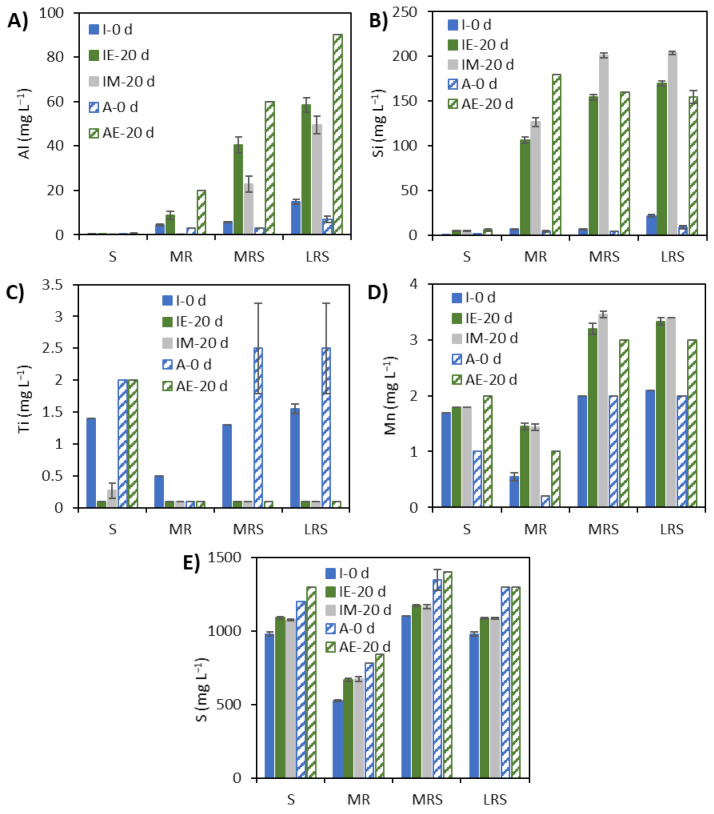
Concentrations of soluble metal cations: (**A**) aluminium, (**B**) silicon, (**C**) titanium, (**D**) manganese and (**E**) sulfur in inoculated (I) and abiotic (A) FPAs after incubation for 0 and 20 days under Earth gravity (E) and simulated microgravity (M) conditions in the presence of 12.5 mM Fe^3+^ (S), 1% Mars regolith simulant (MR), 1% Mars regolith simulant plus 12.5 mM Fe^3+^ (MRS) and 1% lunar regolith simulant plus 12.5 mM Fe^3+^ as substrates (LRS).

**Figure 6 microorganisms-09-02416-f006:**
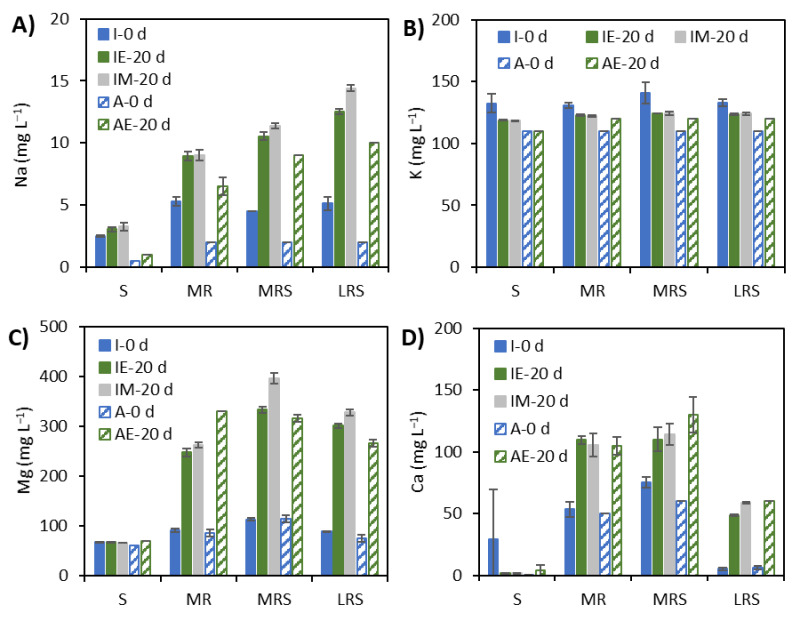
Concentrations of soluble elements: (**A**) sodium, (**B**) potassium, (**C**) magnesium and (**D**) calcium in inoculated (I) and abiotic (A) FPAs after incubation for 0 and 20 days under Earth gravity (E) and simulated microgravity (M) conditions in the presence of 12.5 mM Fe^3+^ (S), 1% Mars regolith simulant (MR), 1% Mars regolith simulant plus 12.5 mM Fe^3+^ (MRS) and 1% lunar regolith simulant 12.5 mM Fe^3+^ (LRS). H_2_/CO_2_ gas mixture was added to all cultures as source of electron donor and carbon source, respectively.

**Figure 7 microorganisms-09-02416-f007:**
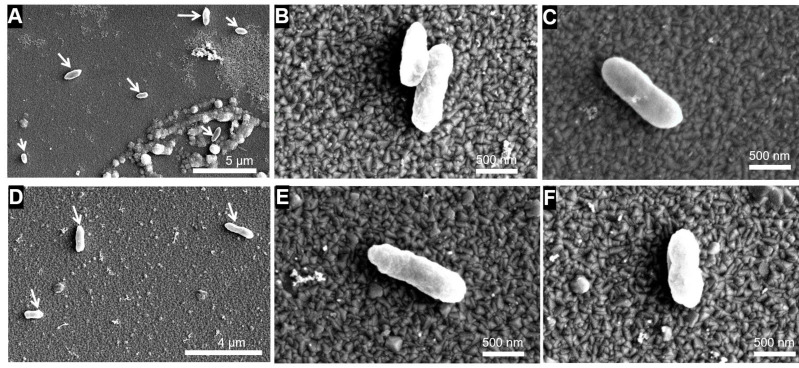
Scanning electron microscopy (SEM) images of *A. ferrooxidans* cells incubated with H_2_/CO_2_ and Fe^3+^ as the electron donor, carbon source and electron acceptor in the absence of regolith simulant under Earth gravity (**A**–**C**) and simulated microgravity (**D**–**F**). White arrows indicate cells.

**Figure 8 microorganisms-09-02416-f008:**
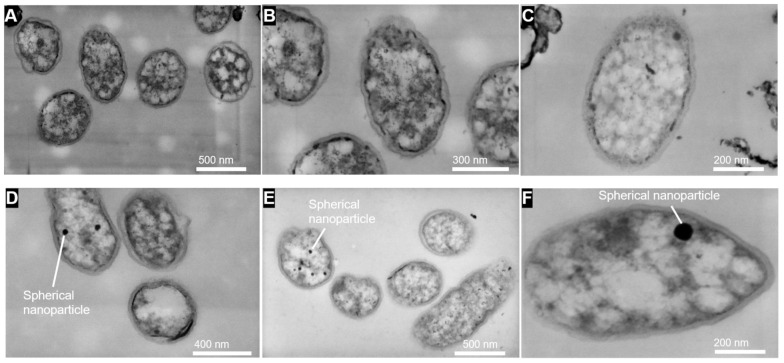
Scanning transmission electron microscopy (STEM) images of 80 nm-thick sections of *A. ferrooxidans* cells incubated with H_2_/CO_2_ and Fe^3+^ as the electron donor, carbon source and electron acceptor in the absence of regolith simulant under Earth gravity (**A**–**C**) and simulated microgravity (**D**–**F**).

**Table 1 microorganisms-09-02416-t001:** Chemical composition of Martian and lunar regolith simulants used in this study (source: ExolithLab).

Major Chemical Composition	Mars Global Simulant (MGS-1)(%-wt)	Lunar Mare Simulant (LMS-1) (%-wt)
SiO_2_	45.57	42.81
TiO_2_	0.30	4.62
Al_2_O_3_	9.43	14.13
Cr_2_O_3_	0.12	0.21
FeO _T_	16.85	7.87
MgO	16.50	18.89
MnO	0.10	0.15
CaO	4.03	5.94
Na_2_O	3.66	4.92
K_2_O	0.43	0.57
P_2_O_5_	0.37	-
SO_3_	2.63	0.11

^T^ = total iron oxide (FeO and Fe_2_O_3_).

**Table 2 microorganisms-09-02416-t002:** Mineral composition of Martian and lunar regolith simulants used in this study (source: ExolithLab).

Mineral Component	Mars Global Simulant (MGS-1)(%-wt)	Lunar Mare Simulant (LMS-1) (%-wt)
Pyroxene	20.3	32.8
Plagioclase	27.1	19.8
Olivine	13.7	11.1
Basalt	-	7.5
Basaltic glass	22.9	-
Ilmenite	-	4.3
Mg-sulfate	4.0	-
Ferrihydrite	3.5	-
Hydrated silica	3.0	-
Magnetite	1.9	-
Anhydrite	1.7	-
Fe-carbonate	1.4	-
Hematite	0.5	-

**Table 3 microorganisms-09-02416-t003:** Experimental conditions for testing metal solubilization from Martian (MR) and lunar regolith simulants (LR) in the presence and absence of substrates (Fe^3+^) (S) and *A. ferrooxidans* inoculum (I) vs. abiotic (A) control under simulated microgravity (M) and/or Earth gravity (E) conditions.

Treatment	Regolith Simulant (1% *w*/*v*)	Electron Acceptor/Donor ^1^	Culture (10% *v/v*)	Gravity
S-IM	-	Fe^3+^ + H_2_/CO_2_	*A. ferrooxidans*	Simulated µg
MR-IM	Martian	H_2_/CO_2_	*A. ferrooxidans*	Simulated µg
MRS-IM	Martian	Fe^3+^ + H_2_/CO_2_	*A. ferrooxidans*	Simulated µg
LRS-IM	Lunar	Fe^3+^ + H_2_/CO_2_	*A. ferrooxidans*	Simulated µg
S-IE	-	Fe^3+^ + H_2_/CO_2_	*A. ferrooxidans*	Earth Gravity
MR-IE	Martian	H_2_/CO_2_	*A. ferrooxidans*	Earth Gravity
MRS-IE	Martian	Fe^3+^ + H_2_/CO_2_	*A. ferrooxidans*	Earth Gravity
LRS-IE	Lunar	Fe^3+^ + H_2_/CO_2_	*A. ferrooxidans*	Earth Gravity
S-AE	-	Fe^3+^ + H_2_/CO_2_	-	Earth Gravity
MR-AE	Martian	H_2_/CO_2_	-	Earth Gravity
MRS-AE	Martian	Fe^3+^ + H_2_/CO_2_	-	Earth Gravity
LRS-AE	Lunar	Fe^3+^ + H_2_/CO_2_	-	Earth Gravity

^1^ Fe^3+^ = electron acceptor, H_2_ = electron donor, CO_2_ = carbon source.

**Table 4 microorganisms-09-02416-t004:** Average ferric consumption rates of *A. ferrooxidans* cultures incubated under Earth gravity and simulated microgravity conditions. S = 12.5 mM Fe^3+^ only, MR = 1% Mars regolith simulant, MRS = 1% Mars regolith simulant plus 12.5 mM Fe^3+^, LR = 1% lunar regolith simulant plus 12.5 mM Fe^3+^.

	Ferric Iron Consumption Rate (mg L^−1^ d^−1^)	
Cultures	Earth Gravity	Simulated Microgravity
S-I	8.3 ± 2.2	15.6 ± 1.1
MR-I	NA	NA
MRS-I	17.1 ± 1.1	22.2 ± 0.2
LRS-I	20.0 ± 1.5	22.3 ± 0.3
S-A	0	ND
MR-A	NA	ND
MRS-A	11.3 ± 3.8	ND
LRS-A	9.9 ± 0.3	ND

NA = not applicable as there was no Fe^3+^ addition to MR cultures. ND = not determined.

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
