# Peer review of "Potential of Acidithiobacillus ferrooxidans to Grow on and Bioleach Metals from Mars and Lunar Regolith Simulants under Simulated Microgravity Conditions"

_microorganisms, 2021, doi:10.3390/microorganisms9122416_

Round 1
Reviewer 1 Report
The presented work deals with the bioleaching of lunar and martian regolith simulants using Acidithibacillus ferooxidans. The Authors demonstrate that biomining model microorganisms could be potentially used for space applications. Microgravity was also shown to stimulate the biosynthesis of intracellular nanoparticles by the bacteria used.
The paper is well written and fits the scope of the Journal and the Special Issue. The methods and techniques used by the Authors are justified. Data are generally logically presented and support conclusions. I have found little confusing only the presentation of bioleaching results, as described below in comments to the Authors.
1. There are some editing errors in the text according to “ferrooxidans”:
(i) Lines 110 and 122 – it should be written in italics
(ii) Line 518 – there is written ”A.ferrooxidans” – a spacebar is missing
2. Section 2.1.
(i) Information is missing on the amount/concentration of bacteria (inoculum) introduced to the medium.
3. Section 2.4: “After the 20-day incubation, samples (1 mL) were filtered through a 0.2 μm syringe 162 filter (Terumo, Philippines) and collected in 1.5 mL tubes.”
(i) The sample volume appears to be relatively small compared to the mentioned analyses. Filtration causes an additional loss of solution. Were the samples diluted?
4. Line 247 (Figure 3) “Cell counts of ferrooxidans inoculated (I) cultures after incubation for 20 days under Earth gravity (E) and simulated microgravity (M) conditions (clinostat) conditions in the presence of 12.5 mM Fe3+ (S);”
(i) repetition to delete
5. Figures 5 and 6 (general comments)
(i) What was the reason for the difference in the initial concentration of a given element in the sample? Theoretically, the initial concentration should be the same if the samples contain the same mineral (martian regolith). Figures suggest that the addition of Fe3+ or bacteria changes the initial concentration of the tested element, for example, Al.
(ii) The graphs in this form are misleading because, e.g. in the case of Ti, K, or Ca, they show that there was more of the element before the process than after (under certain conditions). Perhaps it would be better to present the results as leaching efficiency in %.
6. Line 325: “No titanium leaching was detected under any conditions as the soluble titanium concentrations were lower after 20 days than on day 0 for all regolith simulant containing FPAs (Figure 5C).
(i) These results suggest that precipitation or accumulation by microorganisms occurs. Was the solid tested after the bioleaching process to clarify this issue?
Author Response
Reviewer 1:
The presented work deals with the bioleaching of lunar and martian regolith simulants using Acidithibacillus ferooxidans. The Authors demonstrate that biomining model microorganisms could be potentially used for space applications. Microgravity was also shown to stimulate the biosynthesis of intracellular nanoparticles by the bacteria used.
The paper is well written and fits the scope of the Journal and the Special Issue. The methods and techniques used by the Authors are justified. Data are generally logically presented and support conclusions. I have found little confusing only the presentation of bioleaching results, as described below in comments to the Authors.
We thank the reviewer for the comment and valuable input.
- There are some editing errors in the text according to “ferrooxidans”:
(i) Lines 110 and 122 – it should be written in italics
(ii) Line 518 – there is written ”A.ferrooxidans” – a spacebar is missing
We have changed the typos A.ferrodixans to A. ferrooxidans in the above mentioned section and in the rest of manuscript.
- Section 2.1.
(i) Information is missing on the amount/concentration of bacteria (inoculum) introduced to the medium.
We have added the missing info (10%) into the manuscript.
- Section 2.4: “After the 20-day incubation, samples (1 mL) were filtered through a 0.2 μm syringe 162 filter (Terumo, Philippines) and collected in 1.5 mL tubes.”
(i) The sample volume appears to be relatively small compared to the mentioned analyses. Filtration causes an additional loss of solution. Were the samples diluted?
We didn’t dilute the samples. We did only take 1 mL for the physico chemical analysis (as described in the manuscript). Of the 13.5 mL cultures, 10 mL samples were taken for Rnaseq. But the amount (and the quality) of RNA extracted of from these samples weren’t good enough to do sequencing.
- Line 247(Figure 3) “Cell counts of ferrooxidansinoculated (I) cultures after incubation for 20 days under Earth gravity (E) and simulated microgravity (M) conditions (clinostat) conditions in the presence of 12.5 mM Fe3+ (S);”
(i) repetition to delete
We have deleted the repetition.
- Figures 5 and 6 (general comments)
(i) What was the reason for the difference in the initial concentration of a given element in the sample? Theoretically, the initial concentration should be the same if the samples contain the same mineral (martian regolith). Figures suggest that the addition of Fe3+ or bacteria changes the initial concentration of the tested element, for example, Al.
The initial concentration of a given element in the sample at day 0 were similar in MR and MRS for Al, Si, Cr. The difference concentration in Ti and Mn (in MR vs MRS) was probably due to the low amount of these metals (close to the detection limit in the analysis). The higher amount of S in MRS was due to the addition of S in the MRS culture. The amount of metal in LRS will be different than in MR and MRS due to different metal composition in lunar regolith.
(ii) The graphs in this form are misleading because, e.g. in the case of Ti, K, or Ca, they show that there was more of the element before the process than after (under certain conditions). Perhaps it would be better to present the results as leaching efficiency in %.
We thank the reviewer for his/her valuable input on the graph. We however think that this type of graph (with y-axis = mg/L of metal) will give additional info such as the exact amount of metal being leached. For some data, the % leaching efficiency graph would probably give high value of leaching, yet the exact amount of metal being leached was quite small. Similar type of graph (as we presented) was also used in Kolbl et al., 2017.
- Line 325:“No titanium leaching was detected under any conditions as the soluble titanium concentrations were lower after 20 days than on day 0 for all regolith simulant containing FPAs (Figure 5C).
(i) These results suggest that precipitation or accumulation by microorganisms occurs. Was the solid tested after the bioleaching process to clarify this issue?
We didn’t do any measurement to the solid to clarify this. We looked at the literatures where similar trend of precipitation has occurred and compared our results.
Reviewer 2 Report
Manuscript ID: microorganisms-1456266
Title: Potential of Acidithiobacillus ferrooxidans to grow on and bioleach metals from Mars and Lunar regolith simulants under simulated microgravity conditions
Dear authors congratulations for your interesting paper. I just have some comments, corrections and suggestions that you find below.
Introduction
Line 41. Human presence, I guess.
Line 46-48. This sentence needs a reference.
Line 65. If the sentence that ends here and the next are connected, as I suppose, they need a connection element as colon or beginning with “In them” (for example) and then, put the reference at the end of “have been reported”.
Line 77-81. This is a too long sentence. Try to do two sentences instead using a long subordinate one. Ex. “Shewanella oneidensis is a ….. . This bacterium and A. ferrooxidans have a theoretical feasibility to be used for extracting…” Or others possibilities.
Line 84-85. Do you mean in this study? Indicate it please.
Line 86. Which location do you refer to?
Line 90. Osmorality, pH and temperature are not stressors, they are conditions or physico-chemical parameters that could be stressing depending on their magnitude and organisms.
Line 97-99. This is a too long subject that makes difficult the Reading of the sentence.
Line 106. “to obtain insights on its biomining potential of space resources” is a repetitive idea indicated two lines before, here is not necessary.
Line 271. Table 4, at the end of the sentence.
Line 285. Except for inoculated microgravity samples in MR.
Line 304-313. You have already talk about Fe solubilisation, levels and redox potential in the previous sub-section so I thin this information is unnecessary here but it could be moved to the previous section when talking about redox potential (line 283 and go on).
Line 325. What about microgravity condition?
Line 328-330. Anything remarkable about the identical leaching of IE and IM with S but without MR and vice versa?
Line 352. You have indicated the p-value for other comparisons, could you also put it for this case in order to be consistent throughout the manuscript?
Line 384. What do you mean by “significant differences” in size? Is there some kind of numerical data of measuring size that I missed?
Line 384. Have you measured the thickness? I think it is not indicated in material and methods section.
Line 386-390. This information should be part of the discussion section it is not a result but an explanation of it.
Line 394-401. The whole paragraph should be part of the discussion section. Besides, you should said what magnetosome are before talking about them in line 394. Finally, there is a need for references at least in line 397 when the sentence finishes and in line 398 after “Fe2+/O2. (A good location for this paragraph could be line 488)
Line 417. The word synthetic is repeated.
Line 418. There is a lack of reference here, also in line 420, at the end of the sentence.
Line 420. Why acidophiles? It should be a brief explanation or indicated reason.
Line 435. […microgravity similar to in space]? Or just do you mean liquid environment but in the space? Also, explain why it influence their viability and survival.
Line 496. Conditions are anoxic not anaerobic.
Line 505. Which drug? You should introduce the idea of using properties derived from effects of microgavity in cells in the medicine filed before launch you to talk about the drugs incorporation…I think the whole paragraph is a little bit lacking in idea´s connection in a fluent and logical way.
Line 508. Why and how magnetic nanoparticules reduce side effects?
Line 509. This last sentence is good for conclusions section.
Line 515. Delete leached, it is not necessary and it is repeated in the sentence.
Author Response
Reviewer 2
Manuscript ID: microorganisms-1456266
Title: Potential of Acidithiobacillus ferrooxidans to grow on and bioleach metals from Mars and Lunar regolith simulants under simulated microgravity conditions
Dear authors congratulations for your interesting paper. I just have some comments, corrections and suggestions that you find below.
We thank the reviewer for his/her valuable input.
Introduction
Line 41. Human presence, I guess.
We have added the suggestion into the manuscript.
Line 46-48. This sentence needs a reference.
We have added two references to this sentence.
Line 65. If the sentence that ends here and the next are connected, as I suppose, they need a connection element as colon or beginning with “In them” (for example) and then, put the reference at the end of “have been reported”.
We have connected the two sentences and add the references.
Line 77-81. This is a too long sentence. Try to do two sentences instead using a long subordinate one. Ex. “Shewanella oneidensis is a ….. . This bacterium and A. ferrooxidans have a theoretical feasibility to be used for extracting…” Or others possibilities.
We have split the long sentences into two.
Line 84-85. Do you mean in this study? Indicate it please.
We have added the “ in this study” into the sentence.
Line 86. Which location do you refer to?
We have added Martian to the sentence.
Line 90. Osmorality, pH and temperature are not stressors, they are conditions or physico-chemical parameters that could be stressing depending on their magnitude and organisms.
We have amended the sentence.
Line 97-99. This is a too long subject that makes difficult the Reading of the sentence.
We have revised the subject of the sentence.
Line 106. “to obtain insights on its biomining potential of space resources” is a repetitive idea indicated two lines before, here is not necessary.
We have deleted the phrase from the sentence.
Line 271. Table 4, at the end of the sentence.
We have added Table 4.
Line 285. Except for inoculated microgravity samples in MR.
Inoculated microgravity samples in MR (IM-20d in Figure 4C) showed a lower redox potential compared to the time 0 (I-0d).
Line 304-313. You have already talk about Fe solubilisation, levels and redox potential in the previous sub-section so I thin this information is unnecessary here but it could be moved to the previous section when talking about redox potential (line 283 and go on).
We have moved the paragraph into the previous section where we described the redox potential.
Line 325. What about microgravity condition?
We have described the leaching result in microgravity condition in the previous sentences.
Line 328-330. Anything remarkable about the identical leaching of IE and IM with S but without MR and vice versa?
We have removed the data (and the corresponding Figure and text) on chromium leaching from the manuscript. What looks like identical leaching of IE and IM was due to the chromium concentration was very low, close to the detection limit.
Line 352. You have indicated the p-value for other comparisons, could you also put it for this case in order to be consistent throughout the manuscript?
We have added the P-value into the manuscript.
Line 384. What do you mean by “significant differences” in size? Is there some kind of numerical data of measuring size that I missed?
We originally thought this info (no difference in size) could be inferred from the SEM/STEM figures. However, we have now added numerical data of cell size measurements into supplementary Figure S1.
Line 384. Have you measured the thickness? I think it is not indicated in material and methods section.
We infer the cell thickness from the SEM/STEM figures.
Line 386-390. This information should be part of the discussion section it is not a result but an explanation of it.
We have moved the sentence into the discussion.
Line 394-401. The whole paragraph should be part of the discussion section. Besides, you should said what magnetosome are before talking about them in line 394. Finally, there is a need for references at least in line 397 when the sentence finishes and in line 398 after “Fe2+/O2. (A good location for this paragraph could be line 488)
We have moved the section into the discussion and added the reference.
Line 417. The word synthetic is repeated.
One synthetic word has been deleted.
Line 418. There is a lack of reference here, also in line 420, at the end of the sentence.
We have added references to line 418 and 420.
Line 420. Why acidophiles? It should be a brief explanation or indicated reason.
Additional explanation has been provided in the sentence. A.ferrooxidans, which is an acidophile, can leach minerals in the regolith and form magnetosome, hence can be considered as micro-factories.
Line 435. […microgravity similar to in space]? Or just do you mean liquid environment but in the space? Also, explain why it influence their viability and survival.
We have revised the sentences and provided explanation in the subsequent sentences (e.g. the absence of sedimentation, convection or hydrostatic pressure under microgravity) might influence the extracellular transport, and hence the cellular growth.
Line 496. Conditions are anoxic not anaerobic.
We have changed the anaerobic phrase into anoxic.
Line 505. Which drug? You should introduce the idea of using properties derived from effects of microgavity in cells in the medicine filed before launch you to talk about the drugs incorporation…I think the whole paragraph is a little bit lacking in idea´s connection in a fluent and logical way.
We have added several introduction sentences about magnetosome as drug delivery systems.
Line 508. Why and how magnetic nanoparticules reduce side effects?
We have added a sentence to explain this “The abundance of primary amino groups in the surface of magnetosomes can be functionalised with diverse bioactive molecules (e.g. ligands) to potentially target the affected area and hence reduce the adverse side effects.”
Line 509. This last sentence is good for conclusions section.
We have moved this sentence to the conclusion.
Line 515. Delete leached, it is not necessary and it is repeated in the sentence.
We have deleted the leached phrase.
Reviewer 3 Report
The paper is an interesting research article that is highly relevant to ISRU. There are some grammatical issues throughout, some of which I have highlighted below with suggested changes on how to improve clarity.
I would be interested to see the effects of microgravity on abiotic leaching, especially with the observation of abiotic ferric precipitation, the proposal of increased ferric reduction in microgravity and how the leaching of elements from the regolith alters. Whilst I don't think that this is essential for the publication of the paper, I think that flagging the absence of this abiotic control in the discussion would be useful when assessing the results.
line 40 the recent launch of NASA's maybe better phrasing
61 suggestion of a possible role or suggested hypothesis as opposed to theory.
97 no study has previously measured
99 only the heterotroph
102 the space flight experiment
Table 3 the underlining makes the table less clear
225 sentence could be rewritten to be clearly e.g. "for solely in the presence of the mars regolith simulants, without the addition"
230 would under earth gravity be more correct?
234 is the comma needed
242 didn typo
290 Describe the significance of the increase
391 STEM?
414 through not to the solar system?
417 trophs plural
440 this sentence is not very clear
444 as the microbes are?
480 the gravity applied
482 were obtained
489 is capable
491 favoured under
496 magnetosome
498 lost
Author Response
Reviewer 3
The paper is an interesting research article that is highly relevant to ISRU. There are some grammatical issues throughout, some of which I have highlighted below with suggested changes on how to improve clarity.
I would be interested to see the effects of microgravity on abiotic leaching, especially with the observation of abiotic ferric precipitation, the proposal of increased ferric reduction in microgravity and how the leaching of elements from the regolith alters. Whilst I don't think that this is essential for the publication of the paper, I think that flagging the absence of this abiotic control in the discussion would be useful when assessing the results.
We thank the reviewer for his/her valuable comment. We have clarified the lack of abiotic control in simulated condition into the discussion section.
line 40 the recent launch of NASA's maybe better phrasing
61 suggestion of a possible role or suggested hypothesis as opposed to theory.
97 no study has previously measured
99 only the heterotroph
102 the space flight experiment
Table 3 the underlining makes the table less clear
225 sentence could be rewritten to be clearly e.g. "for solely in the presence of the mars regolith simulants, without the addition"
230 would under earth gravity be more correct?
234 is the comma needed
242 didn typo
290 Describe the significance of the increase
391 STEM?
414 through not to the solar system?
417 trophs plural
440 this sentence is not very clear
444 as the microbes are?
480 the gravity applied
482 were obtained
489 is capable
491 favoured under
496 magnetosome
498 lost
We have amended the grammatical error as indicated by the reviewer in the revised manuscript.